# An Exogenous Pre-Storage Melatonin Alleviates Chilling Injury in Some Mango Fruit Cultivars, by Acting on the Enzymatic and Non-Enzymatic Antioxidant System

**DOI:** 10.3390/antiox11020384

**Published:** 2022-02-14

**Authors:** Renu Bhardwaj, Sunil Pareek, J. Abraham Domínguez-Avila, Gustavo A. Gonzalez-Aguilar, Daniel Valero, Maria Serrano

**Affiliations:** 1Department of Agriculture and Environmental Sciences, National Institute of Food Technology Entrepreneurship and Management, Kundli, Sonepat 131028, Haryana, India; renu.bhardwaj@niftem.ac.in; 2Coordinacion de Tecnología de Alimentos de Origen Vegetal, Centro de Investigación en Alimentación y Desarrollo A.C. (CIAD), Carretera Gustavo Enrique Astiazaran Rosas No. 46, Hermosillo 83304, Mexico; abrahamdominguez9@gmail.com (J.A.D.-A.); gustavo@ciad.mx (G.A.G.-A.); 3Department of Food Technology, Escuela Politécnica Superior de Orihuela, University Miguel Hernández, Ctra. Beniel km. 3.2, 03312 Alicante, Spain; daniel.valero@umh.es; 4Department of Applied Biology, Escuela Politécnica Superior de Orihuela, University Miguel Hernández, Ctra. Beniel km. 3.2, 03312 Alicante, Spain

**Keywords:** antioxidant enzymes, cold storage, *Mangifera indica*, phenolics, phenylalanine ammonia-lyase, quality preservation, ROS

## Abstract

Melatonin (MT) treatment (100 µM, 2 h) was applied to four mango fruit cultivars (‘Langra’, ‘Chaunsa’, ‘Dashehari’, and ‘Gulab Jamun’), before being stored at 5 ± 1 °C for 28 d, in order to alleviate chilling injury (CI). Maximum CI reduction was observed in ‘Langra’ mangoes, and minimum in ‘Gulab Jamun’ mangoes. This positive effect on quality preservation was associated with an increased concentration of endogenous MT, which prevented the accumulation of reactive oxygen species (H_2_O_2_ and O_2_^·−^) and stimulated non-enzymatic antioxidants (total phenolic compounds and total flavonoids), possibly due to higher activity of phenylalanine ammonia lyase and tyrosine ammonia lyase. Increased antioxidant activity was also documented in MT-treated ‘Langra’ mangoes, according to four different assays (DPPH, TEAC, FRAP, and CUPRAC) and higher activity of six antioxidant enzymes (superoxide dismutase, catalase, peroxidase, ascorbate peroxidase, glutathione reductase, and dehydroascorbate reductase). In contrast, ‘Gulab Jamun’ mangoes showed minimal or no positive effects on the aforementioned variables in response to the exogenous MT application. ‘Chaunsa’ and ‘Dashehari’ mangoes had some intermediate effects on their antioxidant system (enzymatic and non-enzymatic) and alleviation of CI, when treated with exogenous MT. We conclude that exogenous MT exerts a cultivar-dependent stimulating effect on the antioxidant system of mangoes, which results in an increase in the fruits’ resistance to low temperature.

## 1. Introduction

Mango (*Mangifera indica* L.) is a tropical fruit, liked across the globe due to its nutritional value and excellent aroma and flavour [1]. It is a climacteric fruit that can be harvested at physiological maturity before ripening. Due to continued evolution of ethylene production and high respiration rates, mangoes take about 9 to 12 d to ripen after harvest, depending on their maturity stage [2]. These processes result in a short shelf life; thus, proper postharvest handling and pre-storage treatments are often necessary to extend their availability period.

Low temperature storage slows down cellular metabolism, and is one of the most common ways to extend the availability of various fruits; however, temperatures <13 °C will result in chilling injury (CI) in tropical fruit such as mango [3]. Fruit that develop CI show some characteristic signs, such as uneven ripening, skin browning, surface pitting, and water-soaked lesions [4]. Therefore, techniques that allow to store mangoes at low temperatures, while also alleviating CI, are useful to extend their shelf life, availability period, and maintenance of export chain.

Numerous approaches have been undertaken to prevent CI in mangoes stored at low temperatures, such as pre-storage applications of methyl jasmonate [5], ethrel [6], oxalic acid and salicylic acid [7], 2,4-dichlorophenoxyacetic acid [8], nitric oxide [9], low temperature conditioning [10], and chitosan and polyamine coating [11]. Although some have shown interesting results, they still have some drawbacks. For example, high concentrations of 2,4-dichlorophenoxyacetic acid are potentially toxic for humans [8], while nitric oxide [9] and chitosan and polyamine coating [10] lead to a reduction in respiration and ethylene production, and thus a poor colour development in mangoes. On contrary, the ripening process was enhanced through application of methyl jasmonate [5], ethrel [6], and low temperature conditioning [11]. Therefore, there is still a need for more natural and reliable methods that can not only enhance cold tolerance of mangoes, but also maintain their overall quality under these conditions.

Melatonin (*N*-acetyl-5-methoxytryptamine, MT) is an amphiphilic, low molecular weight, indolic compound ubiquitous in living organisms [12]. It acts as a bio-stimulant, antioxidant, and powerful regulator of growth and development under biotic and abiotic stress conditions [13]. Postharvest MT applications on banana [14], broccoli [15], apple [16], and strawberry [17] fruit have highlighted its role in quality improvement, delaying senescence, preventing browning, and reducing fungal infection, respectively. MT has also shown a significant effect against CI in peach [18], tomato [19], and pomegranate [20] fruit.

The positive effects of its application are associated with the accumulation of endogenous MT, which acts as an antioxidant [17] by stimulating the free radical scavenging system [21], inducing activity of the pentose phosphate pathway [22], and maintaining the saturated-to-unsaturated ratio of membrane phospholipids at low levels [22]. It has also been shown to inhibit the synthesis of ethylene biosynthesising enzymes [14], promoting the GABA shunt, and regulating intracellular energy production, polyamine, and proline accumulation [18].

We have reported the effects of an exogenous MT application in mangoes in an earlier study, which showed that they are cultivar-dependent, and that its efficacy is closely associated with proline metabolism [4] and γ-aminobutyric acid shunt pathway [23]. Thus, according to the literature and to our previous experiments, we hypothesise that a pre-storage MT application may exert cultivar-dependent effects to alleviate CI signs in mango, which may be associated with changes in the fruits’ enzymatic and non-enzymatic antioxidant system.

The objectives of the present study were to (a) evaluate the effects of pre-storage MT application on CI signs of four cultivars of mango fruit stored at 5 ± 1 °C, (b) evaluate the effects of an exogenous MT on endogenous MT content, and (c) elucidate the relation between CI and enzymatic and non-enzymatic antioxidant metabolism of mango fruit.

## 2. Materials and Methods

### 2.1. Plant Material and Treatment

Four mango fruit cultivars were used in the present study, namely, ‘Langra’, ‘Chaunsa’, ‘Dashehari’ and ‘Gulab Jamun’. Fruits were handpicked at physiological maturity but unripened, from an orchard of a local producer located in Sonepat, Haryana, India, and immediately transported to the laboratory. Once there, fruits with apparent signs of injury or defects were discarded, while the remaining ones were disinfected by dipping them in a 1% (*v*/*v*) sodium hypochlorite solution for 2 min. For each cultivar, 200 disinfected fruits were then randomly divided into two lots of 100 fruit each. The first lot was dipped in distilled water (control group), and the second one in a 100 µM MT solution for 2 h, under low light conditions, at 26 ± 2 °C (treated group). Both MT concentration and dipping duration were optimised with preliminary experiments, where the concentration of 100 µM and dipping time of 2 h were found to be the best combination, and were in agreement with previous studies by Cao et al. [18,21] and Liu et al. [17].

All fruit in the control and treated groups were air-dried for 2 h, and stored at 5 ± 1 °C with 85–90% relative humidity for a total of 28 d. After 0, 7, 14, 21, and 28 d of cold storage 3 samples of 3 fruits (replicates) were taken at random and left for 3 days at 20 °C. Analyses were performed in triplicates, with each replicate consisting of three fruits. Peel and flesh tissue samples were taken for each replicate and then frozen with liquid nitrogen at the time of observation, and stored at −80 °C for subsequent analyses.

### 2.2. Chilling Injury Index (CII)

The assessment of CI index (CII) was conducted according to the method described by Concellón et al. [24], with some modifications. For this, a ranking scale was used according to visual CI signs (greyish scald, pitting, skin discoloration, etc.), where rank 0 denotes none, rank 1 denotes 1 to 20%, rank 2 denotes 21 to 40%, rank 3 denotes 41 to 60%, rank 4 denotes 61 to 80%, and rank 5 denotes 81 to 100% CI signs over the surface of fruit. CII was reported as percent (%), according to the sum of product of rank and number of fruit in that scale, divided by the total number of fruit observed, according to the following formula:
$$\mathrm{CII}\;=\;\frac{\sum \left(rank\;score\;\times \;number\;of\;fruit\;recieving\;in\;each\;CI\;rank\right)}{total\;number\;of\;fruit}$$

### 2.3. Endogenous MT Content

Endogenous MT content was quantified according to Ma et al. [25], with some modification. For this, 0.1 g of fruit tissue (peel or pulp) was ultrasonicated (PHUC-100, Phoenix, Bengaluru, India) with 1 mL methanol, at 45 °C, for 30 min. The ultrasonicated mixture was then centrifuged (3-18KS, Sigma, Ostrode, Germany) at 12,000× *g*, at 4 °C, for 15 min. The supernatant was recovered and used for the determination of endogenous MT content, with an enzyme-linked immunosorbent assay (ELISA) kit (Melatonin ELISA, GenAsia, Shanghai, China), as per the protocol described by the manufacturer.

### 2.4. H_2_O_2_ Content

The H_2_O_2_ content was determined according to the method of Sergiev et al. [26], with some modifications. For this, 1 g of fruit tissue (peel or pulp) was homogenised (IKA T18 Digital Ultra-Turrax, Cole-Parmer, India) in 10 mL of a chilled 0.1% (*v*/*v*) trichloroacetic acid (TCA) solution. The homogenate was centrifuged at 12,000× *g*, at 4 °C, for 15 min, and the supernatant obtained was used to quantify H_2_O_2_. The reaction mixture consisted of 10 mM potassium phosphate buffer (pH 7.0), 1 M KI and the previously obtained supernatant. The reaction was monitored at 390 nm (Specord200plus, AnalytikJena, Jena, Germany), and a standard curve of H_2_O_2_ was used to calculate the content of this molecule in the samples. Results were expressed as nM of fresh weight.

### 2.5. Superoxide Anion (O_2_^·−^) Content

The content of superoxide anion (O_2_^·−^) was determined according to the method of Elstner [27], with some modifications. An extract was prepared using 1 g of fruit tissue (peel or pulp), which was homogenised in 50 mM potassium phosphate buffer (pH 7.8) and centrifuged at 12,000× *g*, at 4 °C, for 20 min. The supernatant was recovered and used for further analysis.

An assay mixture was prepared, which contained 1 M hydroxylammonium chloride and the supernatant. The mixture was incubated for 1 h at 25 °C, and 2 mL of ether was then added to prevent chlorophyll interference. After incubation, the solution was centrifuged at 12,000× *g* for 10 min. The water layer was mixed with 1 mL of 7 mM α-naphthylamine (3:1, *v*/*v* solution in glacial acetic acid/water) and 17 mM *p*-aminophenylsulphonic acid (3:1, *v*/*v* solution in glacial acetic acid/water). The mixture was incubated for 20 min at 25 °C, followed by immediate measurement of its absorbance at 530 nm. A sodium nitrate curve was used to calculate O_2_^·−^ production, which was expressed as µM fresh weight.

### 2.6. Total Phenolic Content (TPC)

The total phenolic content (TPC) was quantified according to the method of Singleton and Rossi [28], with some modifications. Extraction was performed from 1 g of fruit tissue (peel or pulp), which was homogenised in 10 mL of 80% (*v*/*v*) ethanol, followed by centrifugation at 10,000× *g*, at 4 °C, for 15 min. The supernatant was recovered, and 100 µL were added to the reaction mixture, which contained 1 mL of 10% (*v*/*v*) Folin–Ciocalteu reagent and 15% (*v*/*v*) sodium carbonate. After a 90 min incubation at room temperature, its absorbance was recorded at 765 nm. Results were calculated with a gallic acid standard curve, and expressed as mg gallic acid equivalents (GAE) kg^−1^ of fresh weight.

### 2.7. Total Flavonoid Content (TFC)

The TFC was quantified in the same extract described for TPC, according to the method of Chang et al. [29] with some modifications. For this, 100 µL of 1 M potassium acetate, 10% (*w*/*v*) of aluminium chloride and crude extract was made up to 3 mL with distilled water in an assay tube. After a 45 min incubation under dark conditions at room temperature, the solution’s absorbance was recorded at 510 nm. Results were expressed as g of quercetin equivalents (QE) kg^−1^ fresh weight.

### 2.8. Extracts for Antioxidant Activity

An extract was prepared and used to measure antioxidant activity. For this, 1 g of fruit tissue (peel or pulp) was homogenised in 10 mL of 80% (*v*/*v*) methanol. The homogenate was centrifuged at 12,000× *g*, at 4 °C, for 20 min. The analyses of the free hydrophilic antioxidant fraction were conducted using the 2,2-diphenyl picrylhydrazyl (DPPH) and Trolox equivalent antioxidant capacity (TEAC) methods, whereas the bound antioxidant fraction was analysed with the ferric reducing antioxidant power (FRAP) and cupric reducing antioxidant power (CUPRAC) methods.

#### 2.8.1. Scavenging Activity of the Free Antioxidant Fraction

##### 2,2-Diphenyl Picrylhydrazyl (DPPH)

The scavenging ability of antioxidants towards the stable DPPH radical was measured according to the method of Brand-Williams et al. [30], with some modifications. The reaction mixture contained 3.9 mL of DPPH solution made in 95% (*v*/*v*) methanol, and 0.1 mL of the extract. The solution was incubated for 30 min, and its absorbance was then measured at 515 nm. The absorbances of the control (with no sample added) and samples were used to calculate the percentage of DPPH radical inhibition, according to the following formula:
$$Radical\;inhibition\;\left(\%\right)=\left(\frac{\left(Abs_{control}-Abs_{sample}\right)}{Abs_{control}}\right)\times 100$$

##### Trolox Equivalent Antioxidant Capacity (TEAC)

The TEAC assay was performed with the 2,2′-azino-bis(3-ethylbenzothialozine-6-sulphonic acid) (ABTS) radical (ABTS*^+^), according to the method reported by Re et al. [31], with some modifications. A 7 mM ABTS*^+^ radical solution was initially prepared using an oxidising agent (2.45 mM potassium persulphate). The reaction mixture consisted of the extract and ABTS*^+^ radical solution, which was then incubated for 10 min at 30 °C. Its absorbance was recorded at 734 nm after the incubation period, and the data used to calculate its scavenging activity, according to the formula described for DPPH. Data is reported as percentage of ABTS radical inhibition.

#### 2.8.2. Antioxidant Activity of the Bound Antioxidant Fraction

##### Ferric Reducing Antioxidant Power (FRAP)

The FRAP assay was measured, according to the method of Benzie and Strain [32], with some modifications. FRAP reagent was prepared by mixing 300 mM acetate buffer (pH 3.6), 10 mM 2,4,6-Tris(2-pyridyl)-s-triazine (TPTZ) in 40 mM HCl and 20 mM FeCl_3_ in a 10:1:1 (*v*/*v*/*v*) ratio. The assay mixture contained 2.9 mL of FRAP reagent and 0.1 mL of extract, which were incubated for 30 min at 37 °C. Its absorbance was read at 593 nm after the incubation period, during which an intense blue colour develops as the ferric-tripyridyltriazine complex is reduced to the ferrous state. Absorbances were used to calculate antioxidant potential and were expressed as g of Trolox equivalents (TE) kg^−1^ fresh weight.

##### Cupric Reducing Antioxidant Power (CUPRAC)

The CUPRAC assay was performed according to the method of Apak et al. [33], with some modifications. The reaction mixture contained of cupric chloride (1.0 × 10^−2^ M) solution, ammonium acetate buffer (1 M, pH 7.0), neocuproine alcoholic solution (7.5× 10^−3^ M) and the extract. The mixture was incubated for 30 min, and its absorbance was recorded thereafter at 450 nm. Results were expressed as g of TE kg^−1^ fresh weight, as calculated with a Trolox standard curve.

### 2.9. Extracts for Phenylalanine Ammonia Lyase (PAL) and Tyrosine Ammonia Lyase (TAL)

Both phenylalanine ammonia lyase (PAL, EC 4.3.1.5) and tyrosine ammonia lyase (TAL, EC 4.3.1.25) were assayed, according to the protocol reported by Khan et al. [34], with some modifications. An extract was initially performed, where 1 g of mango fruit tissue (peel or pulp) was homogenised in ice-cooled Tris-HCl buffer (50 mM, pH 8.5) containing 5% (*w*/*v*) polyvinylpolypyrrolidone (PVPP) and 14.4 mM β-mercaptoethanol. The mixture was centrifuged at 10,000× *g*, at 4 °C, for 20 min, and the supernatant recovered was used to quantify PAL and TAL activities.

#### 2.9.1. Phenylalanine Ammonia Lyase (PAL)

The reaction mixture to quantify PAL activity consisted of 800 µL of Tris-HCl buffer (0.5 mM, pH 8.0), 600 mM of l-phenylalanine, and 100 µL of enzyme extract. The mixture was incubated for 1 h at 40 °C, and 100 µL of 5 N HCl was then added to stop the reaction. The absorbance of this solution was read at 290 nm and used to calculate enzyme activity. Results were expressed as U kg^−1^ of protein, where a unit of enzyme activity was defined as the amount of enzyme that produces 1 nmole of cinnamic acid min^−1^.

#### 2.9.2. Tyrosine Ammonia Lyase (TAL)

The reaction mixture to quantify TAL activity contained 800 µL of Tris-HCl buffer (0.5 mM, pH 8.0), 100 µL of 5.5 µM of l-tyrosine, and 100 µL of enzyme extract. The mixture was incubated for 1 h at 40 °C, and 100 µL of 5 N HCl were then added to stop the reaction. The absorbance of this solution was read at 333 nm and used to calculate enzyme activity. Results were expressed as U kg^−1^ of protein, where a unit of enzyme activity was defined as the amount of enzyme that produces 1 nmole of coumaric acid min^−1^.

### 2.10. Extracts for Superoxide Dismutase (SOD), Catalase (CAT), Peroxidase (POX), Ascorbate Peroxidase (APX), Glutathione Reductase (GR) and Dehydroascorbate Reductase (DHAR)

An extract was initially prepared by homogenising 1 g of fruit tissue (peel or pulp) in pre-cooled 0.1 M potassium phosphate buffer (pH 7.0). The mixture was centrifuged at 12,000× *g*, at 4 °C, for 20 min, and the supernatant recovered was used to quantify the activity of the antioxidant enzymes, namely, superoxide dismutase (SOD), catalase (CAT), peroxidase (POX), ascorbate peroxidase (APX), glutathione reductase (GR) and dehydroascorbate reductase (DHAR).

#### 2.10.1. Superoxide Dismutase (SOD)

The activity of SOD (EC 1.15.1.1) was quantified according to the nitroblue tetrazolium (NBT) reduction method described by Kono [35], with some modifications. The reaction mixture contained 50 mM sodium carbonate buffer (pH 10), 0.1 M NBT, and 0.6% (*v*/*v*) triton-X. The reaction was initiated by adding 20 mM hydroxylamine hydrochloride (pH 6.0), and its absorbance was recorded at 540 nm after 2 min of incubation after the enzyme extract was added. A unit of SOD activity was defined as the amount of enzyme required to inhibit chromogen production by 50%, and expressed as U kg^−1^ of protein.

#### 2.10.2. Catalase (CAT)

The activity of CAT (EC 1.11.1.6) was quantified according to the H_2_O_2_ decomposition method reported by Aebi [36], with some modifications. The reaction mixture contained 0.1 M phosphate buffer (pH 7.0) and 150 mM H_2_O_2_. The reaction was started by adding the enzyme extract, and was continuously monitored for 1 min at 240 nm. A unit of CAT activity was defined as the amount of enzyme required to catalyse half of the H_2_O_2_ present in the solution, and was expressed as U kg^−1^ of protein, using 6.3 × 10^−3^ mM^−1^ cm^−1^ as the extinction coefficient.

#### 2.10.3. Peroxidase (POD)

The activity of POD (EC 1.11.1.7) was quantified by measuring the production of an oxidised compound using guaiacol as substrate, according to the method reported by Putter [37], with some modifications. The reaction mixture contained 0.1 M phosphate buffer (pH 7.0), 20 mM guaiacol, 12.3 mM H_2_O_2_, and the enzyme extract. The reaction was monitored at 436 nm for the formation of guaiacol dehydrogenation product (GDHP) as end product. A unit of POD activity was defined as the amount of enzyme required to produce 1.0 µmol GDHP min^−1^. The activity of POD was expressed as U kg^−1^ of protein, using 25 mM^−1^ cm^−1^ as the extinction coefficient.

#### 2.10.4. Ascorbate Peroxidase (APX)

The activity of APX (EC 1.11.1.11) was quantified according to the ascorbate oxidation method reported by Nakano and Asada [38], with some modifications. The reaction mixture contained 0.1 M phosphate buffer (pH 7.0), 5 mM ascorbate, 0.5 mM H_2_O_2_, and the enzyme extract. The reaction was monitored at 290 nm for 1 min, and the change in absorbance was used to quantify the activity of APX. Results were expressed as U kg^−1^ of protein, using 2.8 mM^−1^ cm^−1^ as extinction coefficient, where a unit of APX activity was defined as the amount of enzyme required to produce 1 nmol of oxidised ascorbate min^−1^.

#### 2.10.5. Glutathione Reductase (GR)

The activity of GR (EC 1.6.4.2) was quantified according to the oxidation of NADPH reported by Calberg and Mannervik [39], with some modifications. The reaction mixture contained 50 mM phosphate buffer (pH 7.0), 3 mM ethylene diamine tetraacetic acid (EDTA) disodium salt, 0.1 mM NADPH, 1 mM oxidised glutathione (GSSG), and the enzyme extract. The reaction was monitored at 340 nm for 1 min, and the change in absorbance was used to quantify the activity of GR. Results were expressed as U kg^−1^ of protein, using 6.22 mM^−1^ cm^−1^ as extinction coefficient, where a unit of GR activity was defined as the amount of enzyme that catalyses the oxidation of 1 nmol of NADPH min^−1^.

#### 2.10.6. Dehydroascorbate Reductase (DHAR)

The activity of DHAR (EC 1.8.5.1) was quantified according to the dehydroascorbate reduction method reported by Dalton et al. [40], with some modifications. The reaction mixture contained 0.1 M phosphate buffer (pH 7.0), 1 mM EDTA, 15 mM reduced glutathione (GSH), 2 mM dehydroascorbate, and the enzyme extract. The reaction was monitored at 265 nm for 1 min. The change in absorbance was used to quantify the activity of DHAR. Results were expressed as U kg^−1^ of protein, using 14 mM^−1^ cm^−1^ as extinction coefficient, where a unit of DHAR activity was defined as the amount of enzyme required to form 1.0 µmol of reduced ascorbate min^−1^.

### 2.11. Protein Concentration

Protein concentration of the extract was determined according to the Bradford [41] method, with some modifications. For this, 1 g of fruit tissue (peel or pulp) was homogenised in 10 mL of chilled 50 mM sodium phosphate buffer containing polyvinylpolypyrrolidone (PVPP), 0.5 mM magnesium chloride, and 2 mM phenylmethylsulphonyl fluoride (PMSF). The mixture was centrifuged at 12,000× *g*, at 4 °C, for 15 min, and the obtained supernatant was used as crude extract. For the analysis, 5 mL of Bradford’s reagent were added to 0.1 mL of crude extract, and its absorbance was read at 595 nm, after 5 min of incubation. Absorbance data was used to calculate protein concentration, by using a calibration curve of bovine serum albumin (BSA) as standard protein.

### 2.12. Statistical Analyses

All experiments described were performed in triplicate, using a completely randomized design (CRD). A one-way analysis of variance (ANOVA) and Duncan’s test were used to determine differences between treatments. Analyses were performed in SPSS (Version 20), with a 0.05 level of significance.

## 3. Results

### 3.1. Chilling Injury Index (CII)

Most fruits treated with MT showed significant (*p* < 0.05) reduction in CI signs, except for the ‘Gulab Jamun’ cultivar (Figure 1a). Treated ‘Langra’, ‘Chaunsa’, ‘Dashehari’, and ‘Gulab Jamun’ mangoes had 4.8, 1.8, 1.7, and 1.1 times less CII severity, respectively, after 28 d of cold storage at 5 ± 1 °C. It is therefore apparent that ‘Langra’ mangoes showed maximum tolerance to CI in response to the application of MT, followed by ‘Chaunsa’ and ‘Dashehari’ mangoes.

### 3.2. Endogenous Melatonin (MT) Content

The exogenous MT had a significant influence on the endogenous concentration of this compound in the peel and pulp of most cultivars, except for ‘Gulab Jamun’ (Figure 1b,c). This is in line with the inference made for CII mentioned in the previous section, where ‘Gulab Jamun’ mangoes had a minimal-to-null response to the application of MT.

### 3.3. O_2_^·−^ and H_2_O_2_ Content

O_2_^·−^ and H_2_O_2_ contents increased in peel and pulp tissues during storage in the four mango cultivars. MT-treated ‘Langra’ mangoes had a significantly lower (*p* < 0.05) O_2_^·−^ content, in both peel (Figure 2a) and pulp (Figure 2b). Specifically, the peel and pulp had five and two times lower values, respectively, as compared to the controls at 21 d of observation. Similarly, the peel of MT-treated ‘Chaunsa’ mangoes had two times lower O_2_^·−^ content at 21 d of storage, as compared to the control, while no significant differences were observed in the flesh (Figure 2c). ‘Dashehari’ and ‘Gulab Jamun’ mangoes did not show any significant response (*p* > 0.05) to the treatment in either tissue (Figure 2e,h). In contrast, the peel of MT-treated ‘Gulab Jamun’ mangoes had a higher O_2_^·−^ content throughout the observation period (Figure 2g).

The effect of MT treatment on H_2_O_2_ content was significant only for the ‘Langra’ cultivar, in which lower values of H_2_O_2_ were observed in the pulp during the whole storage. However, for the remaining cultivars the peel and pulp of mangoes did not show any significant response to the MT treatment, regarding their H_2_O_2_ content (Figure 3c,d).

### 3.4. Total Phenolic Content (TPC) and Total Flavonoid Content (TFC)

In order to determine the influence of MT on secondary metabolites, namely, phenolics and flavonoids, they were quantified in both the peel and pulp. In the peel (Figure 4a), TPC was higher in response to MT treatment in ‘Langra’, ‘Chaunsa’, and ‘Dashehari’ mangoes, while the opposite was found in the peel of ‘Gulab Jamun’ mangoes. In the pulp (Figure 4b), TPC of the treated groups was similar to that of the controls, although there is a tendency to a decrease in response to the treatment in ‘Gulab Jamun’ mangoes.

The TFC of peel was significantly higher in MT-treated fruit at the 28 d of observation in ‘Langra’ mangoes, and all observed days in ‘Chaunsa’ mangoes (Figure 4c). In contrast, no statistically significant effect was apparent on MT-treated ‘Dashehari’ or ‘Gulab Jamun’ mangoes on 21 and 28 d of observation. In the pulp of MT-treated mangoes (Figure 4d), TFC increased on some days of observation in ‘Langra’ and ‘Chaunsa’ mangoes, no effect was found on ‘Dashehari’ mangoes, and a decrease on some days was found on ‘Gulab Jamun’ mangoes.

According to this data, it is apparent that the effect of the MT treatment on TPC and TFC was cultivar- and tissue-dependent. Some significant responses were apparent on ‘Langra’, ‘Chaunsa’ and ‘Dashehari’ mangoes, with minimal-to-null significant responses on ‘Gulab Jamun’ mangoes.

### 3.5. Radical Scavenging Activity (DPPH, TEAC, FRAP, and CUPRAC)

MT treatment induced a higher DPPH activity in both the peel (Figure 5a) and pulp (Figure 5b) of ‘Langra’ mangoes on some days of observation. The peel (Figure 5c), but not the pulp (Figure 5d), of MT-treated ‘Chaunsa’ mangoes had a significantly higher DPPH value at 7 d of observation. No statistically significant effects were apparent on the DPPH values of either the peel or pulp of ‘Dashehari’ (Figure 5e,f) or ‘Gulab Jamun’ (Figure 5g,h) mangoes.

The peel of MT-treated ‘Langra’ (Figure 6a), ‘Chaunsa’ (Figure 6c), and ‘Dashehari’ (Figure 6e) mangoes had significantly higher TEAC values, on at least one day of observation. In contrast, the peel of MT-treated ‘Gulab Jamun’ mangoes had a significantly lower TEAC value at 14 d of observation (Figure 6g). In the case of pulp of most MT-treated mangoes (Figure 6b,d,f), except for ‘Gulab Jamun’ (Figure 6h), no significant differences were found (*p* > 0.05), although a tendency towards increasing is apparent.

FRAP values for the peel and pulp of MT-treated ‘Langra’ (Figure 7a,b) and ‘Chaunsa’ (Figure 7c,d) mangoes were significantly higher (*p* < 0.05), while no effect was apparent on ‘Dashehari’ (Figure 7e,f) and ‘Gulab Jamun’ (Figure 7g,h) mangoes (*p* > 0.05). Most MT-treated mangoes had higher CUPRAC values in both the peel and pulp (Figure 8), with the exception of the pulp of ‘Gulab Jamun’ mangoes (Figure 8h).

According to these findings, it is apparent that each method to quantify antioxidant capacity yielded different behaviours and responded differently to each tissue or cultivar. For example, changes in DPPH and TEAC were with tissue-dependent, changes in FRAP were cultivar-dependent, while all tissues and cultivars exerted changes in CUPRAC values.

### 3.6. Activity of Enzymes Related to the Antioxidant Metabolism

#### 3.6.1. Phenylalanine Ammonia-Lyase (PAL) and Tyrosine Ammonia-Lyase (TAL)

The enzymes PAL and TAL are both part of the phenylpropanoid pathway. The peel of MT-treated ‘Langra’ mangoes had a significantly higher (*p* < 0.05) PAL activity at 21 d of observation (Figure 9a). MT-treated ‘Chaunsa’ mangoes showed a tendency to increase this enzyme’s activity, but it was only apparent in its peel (Figure 9c). 

The activity of TAL increased significantly (*p* < 0.05) in the peel and pulp of MT-treated ‘Langra’ (Figure 10a,b), ‘Chaunsa’ (Figure 10c,d), and ‘Dashehari’ (Figure 10e,f) mangoes, on at least one day of observation. No effect was apparent (*p* > 0.05) on either the peel or pulp of MT-treated ‘Gulab Jamun’ mangoes (Figure 10g,h).

According to this data, TAL activity remained mostly unchanged in all studied cultivars, tissues and times, while PAL activity increased in response to the MT treatment, mainly in the peel of ‘Langra’ mangoes.

#### 3.6.2. Superoxide Dismutase (SOD)

SOD activity increased significantly in the peel and pulp of MT-treated ‘Langra’ (Figure 11a,b) and ‘Chaunsa’ (Figure 11c,d) mangoes, on at least one day of observation, while no effect was apparent in the peel (Figure 11g) or pulp (Figure 11h) of MT-treated ‘Gulab Jamun’ mangoes.

#### 3.6.3. Catalase (CAT)

MT treatment induced a significant increase (*p* < 0.05) on CAT activity in both peel (Figure 12a) and pulp (Figure 12b) of ‘Langra’ mangoes, on at least one day of observation. MT-treated ‘Chaunsa’ mangoes had higher (*p* < 0.05) CAT activity in the peel (Figure 12c) at 21 d of observation, while a tendency to increase was apparent in its pulp (Figure 12d). CAT activity was significantly higher (*p* < 0.05) in the peel (Figure 12e) of ‘Dashehari’ mangoes, while no effect (*p* > 0.05) was apparent on its pulp (Figure 12f). A tendency to increase the activity of CAT was apparent in the peel and pulp of ‘Gulab Jamun’ mangoes (Figure 12g,h).

#### 3.6.4. Peroxidase (POD)

POD activity was significantly higher in MT-treated peel and pulp of ‘Langra’ mangoes at 28 d (Figure 13a) and 21 d (Figure 13b) of observation, respectively. The peel of MT-treated ‘Chaunsa’ mangoes also had higher POD activity at 21 d (Figure 13c) of observation, with no significant effects on its pulp (Figure 13d). MT-treated ‘Dashehari’ mangoes had higher (*p* < 0.05) activity of this enzyme on their peel (Figure 13e) and pulp (Figure 13f), at 14 d and 7 d of observation, respectively. A tendency to increase the activity of this enzyme was apparent on MT-treated ‘Gulab Jamun’ mangoes in both its peel (Figure 13g) and pulp (Figure 13h), although the effect was not significant.

According to these data, the activities of SOD, CAT and POD increased in response to the MT treatments, mainly in the peel of ‘Langra’, ‘Chaunsa’ and ‘Dashehari’ mangoes, with no effects on ‘Gulab Jamun’ mangoes.

#### 3.6.5. Ascorbate Peroxidase (APX)

APX activity of MT-treated ‘Langra’ mangoes was higher at 21 d of storage in both their peel (Figure 14a) and pulp (Figure 14b). MT-treated ‘Chaunsa’ mangoes also had higher APX activity in their peel (Figure 14c) and pulp (Figure 14d), at 21 d and 7 d of observation, respectively. The peel (Figure 14e) of MT-treated ‘Dashehari’ mangoes had a higher APX activity at 14 d of observation. A tendency to increase the activity of APX was found at 21 d of observation in the peel (Figure 14g) of MT-treated ‘Gulab Jamun’ mangoes, but no effect or tendency was found in the pulp.

#### 3.6.6. Glutathione Reductase (GR)

The GR activity of MT-treated mangoes tended to increase in all groups but found to be non-significant (data shown in Appendix A). 

#### 3.6.7. Dehydroascorbate Reductase (DHAR)

MT treatment appeared to induce some tendencies to increase the activity of DHAR, for example, in the peel and pulp of most mangoes, with the exception of the pulp of ‘Gulab Jamun’ mangoes. However, this was found to be non-significant (data shown in Appendix A).

## 4. Discussion

Storing mangoes under low temperature conditions may serve to extend their market availability, but it is well known that storing them at <13 °C results in signs of CI, which restrict their marketability [3]. Previous studies have shown that an application of 100 µM MT in peaches [22], tomatoes [19], and pomegranates [20] induced cold tolerance and alleviated the severity of CI signs. In accordance with these papers, the present study shows alleviation of CI signs in response to a 100 µM, 2 h MT application. Furthermore, the effects appear to be cultivar-dependent, with maximum and minimum efficacy observed in ‘Langra’ and ‘Gulab Jamun’ mangoes, respectively, with ‘Chaunsa’ and ‘Dashehari’ mangoes showing intermediate responses to the treatment.

The aforementioned variability in response to the MT treatment may be due to changes exerted at the physiological, hormonal, and molecular level, through mechanisms that regulate CI (Figure 15). Some of the most crucial factors include molecules collectively known as reactive oxygen species (ROS). These tend to accumulate and damage the cells, due to a disproportionate increase in their production due to chilling stress, in comparison to the activities of the antioxidant defence mechanisms [42]. The oxidative stress due to excessive ROS concentration is potent enough to disturb the metabolism of postharvest produce [43]. As shown in Figure 1, Figure 2 and Figure 3, and in addition to senescence, CI progresses with increasing storage time, together promoting the accumulation and consequent damage exerted by ROS (O_2_^·−^, HO^−^ and H_2_O_2_, among others). Oxidative stress is particularly aggressive on the cell membrane structure and function due to lipid peroxidation, inhibition of enzymatic activities, oxidation of protein, and DNA and RNA damage [44].

The accumulation of endogenous MT in flesh tissue, particularly during the initial days of storage, may be due to an increase in endogenous MT content, due to the upregulation of MT biosynthesis genes, namely, *T5H, ASMT, TDC, SANT,* and *COMT* [45]. Tijero et al. [46] and others [47,48] report that oxidative stress produced during abnormal circumstances (and in addition to exogenous MT application), may upregulate MT biosynthetic genes, thereby culminating in a higher concentration of endogenous MT. The data reported in the present study agrees with these papers, where high endogenous MT, in addition to low O_2_^·−^ (Figure 2), H_2_O_2_ (Figure 3), and MDA [4], resulted in maintenance of membrane stability [17,24]. The protective effect is particularly noticeable in ‘Langra’ mangoes, followed by ‘Chaunsa’ and ‘Dashehari’ mangoes. In contrast, the effects of the MT treatment were mostly non-significant on CII, endogenous MT content (Figure 1), O_2_^·−^ (Figure 2), and H_2_O_2_ content (Figure 3) in ‘Gulab Jamun’ mangoes.

High endogenous MT content after MT treatment could lead to membrane integrity maintenance in most mango cultivars, except for ‘Gulab Jamun’. Tan et al. [49] and Galano et al. [50] established the relationship between the antioxidant activity of MT against ROS and reactive nitrogen species (RNS), in addition to acting as a signalling molecule when the fruit were subjected to stress, similar findings to the ones reported in the present work. The antioxidant activity of MT has been reported to be due to different mechanisms of action, as follows: (1) by directly scavenging ROS and RNS, (2) by stimulating the activity of antioxidant enzymes, (3) by maintaining the non-enzymatic antioxidants in a reduced state, and (4) by preventing ROS formation on the mitochondrial electron transport chain (ETC). These complementary mechanisms collectively prevent ROS and RNS formation, and/or mitigate their oxidative damage [49,50,51].

The nutritive content of mango fruit has been ascribed to its high phenolic and flavonoid content [52]. Both of them are secondary metabolites, which not only provide colour, astringency, bitterness, and a rich flavour to the fruit, but also impart antioxidant properties to it by augmenting its response to oxidative stress [53]. Its phenolic content is maintained or increased in MT-treated mangoes, through upregulation of gene and protein expression of PAL and TAL, both of which are key enzymes involved in the biosynthesis of phenolics and flavonoids through the shikimate-phenylpropanoid-flavonoids pathway [54,55]. The activity of the pathway is directly responsible for the production of various structural and defence-related phenolics such as lignins, phenolic acids, flavonoids, and stilbenes [56,57]. The higher TPC and TFC content in peel (Figure 4a,c) and pulp (Figure 4b,d) of MT-treated ‘Langra’ and ‘Chaunsa’ mangoes, may be associated with higher PAL (Figure 9) and TAL (Figure 10) activities. This observation is in line with previous studies conducted with strawberry [17], litchi [16], and mango [52].

The antioxidant capacity reported in the present study was quantified through four assays, namely, DPPH, TEAC, FRAP, and CUPRAC. Scavenging capacity against the DPPH and ABTSꞏ+ radicals has been positively correlated with TPC and TFC, in MT-treated strawberry fruit [17]. Accordingly, we propose that the reason behind higher antioxidant capacity in each assay of MT-treated ‘Langra’ mangoes, is due to a higher content of TPC and TFC. In contrast, the minor or negligible effects exerted by the MT treatment on ‘Gulab Jamun’ mangoes on their TPC and TFC content, were also reflected on non-significant changes on the antioxidant capacity assays. However, ‘Chaunsa’ and ‘Dashehari’ mangoes showed intermediate effects on their TPC and TFC values in response to the MT treatment, and likewise on their antioxidant capacity, thereby supporting the argument for a cultivar-dependent response.

The MT treatment exerted some positive and significant effects on the activity of antioxidant enzymes (SOD, CAT, POD), most notably, in the peel and pulp of ‘Langra’ mangoes. In contrast, the effects were non-significant or negative in ‘Gulab Jamun’ mangoes, with ‘Chaunsa’ and ‘Dashehari’ mangoes once again showing an intermediate response.

The increased activity of antioxidant enzymes in response to the MT treatment is in accordance with the redox network model [47], where they induce O_2_^·−^ and H_2_O_2_ elimination and reduction in their concentration. Other studies have been performed on eggplants [58], pears [59], and strawberries 17], where an exogenous MT also resulted in clearing ROS, by enhancing the gene expression of antioxidant enzymes.

Ascorbic acid (AsA) and GSH are other important elements to control ROS concentration, according to their role in the AsA-GSH cycle, which aims to counter oxidative stress [60]. Enzymes such as APX, DHAR, GR, and monodehydroascorbate reductase (MDHAR), are also critically involved in the AsA-GSH cycle. The content of AsA is regulated through its synthesis by APX, and is recycled by DHAR and MDHAR. DHAR oxidises GSH to GSSH to mitigate oxidative stress, while subsequent reduction to GSH is performed by GR, in order to preserve a pool of this compound in its reduced form [61]. APX is a pivotal enzyme in chloroplast H_2_O_2_ detoxification through the formation of AsA. Its activity increased in response to an exogenous MT application, in both the peel and pulp of ‘Langra’ and ‘Chaunsa’ mangoes. The decrease in O_2_^−^ and H_2_O_2_ concentration reported in the present study resulted in an increase in AsA and GSH concentration, possibly due to a concomitant increase in GR activity. Although this was only evident in the peel and pulp of ‘Langra’ mangoes, whereas the other cultivars only showed a significant response to the treatment on GR activity in the peel. Finally, DHAR activity was higher in the peel and pulp of most MT-treated cultivars, with the exception of ‘Gulab Jamun’. The influence of MT on enzymes related to the AsA-GSH cycle was in accordance with the study conducted by Tan et al. [49], performed on Chinese flowering cabbage.

## 5. Conclusions

Pre-storage MT (100 µM, 2 h) was applied to four mango cultivars, which were then stored at 5 ± 1 °C for 28 d, in order to determine its effect on alleviation of CI. ‘Langra’ mangoes showed maximum response to the treatment and, therefore, had a higher CI alleviation. The increased chilling tolerance in response to the MT application was attributed to a higher endogenous MT accumulation, which then stimulated the enzymatic and non-enzymatic antioxidant system. The resulting enhanced antioxidant system appears to have maintained cellular homeostasis by decreasing ROS concentration, which increased the fruits’ stress tolerance. This was not apparent in ‘Gulab Jamun’ mangoes, which responded minimally or not at all to the MT treatment, and were therefore unable to develop chilling tolerance as a consequence of MT treatment. ‘Chaunsa’ and ‘Dashehari’ mangoes showed an intermediate response to the treatment and, likewise, a response to CI that was between the one described for ‘Langra’ and ‘Gulab Jamun’ mangoes. This was due to a moderate response of their enzymatic and non-enzymatic antioxidant system.

## Figures and Tables

**Figure 1 antioxidants-11-00384-f001:**
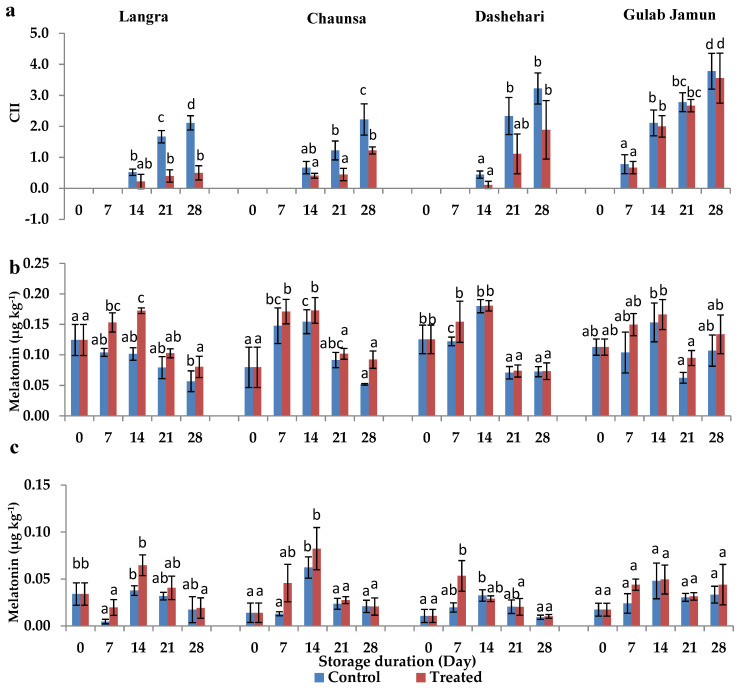
(**a**) Chilling injury index (CII), (**b**) endogenous melatonin (MT) content in peel and (**c**) pulp of ‘Langra’, ‘Chaunsa’, ‘Dashehari’ and ‘Gulab Jamun’ mangoes, as affected by 100 µM melatonin (MT) for 2 h during storage at 5 ± 1 °C plus 3 d of shelf life at room temperature. Data are the mean of three replicates ± standard error. Different lowercase letters indicate significant differences between control and treated fruit for each sampling date (*p* < 0.05).

**Figure 2 antioxidants-11-00384-f002:**
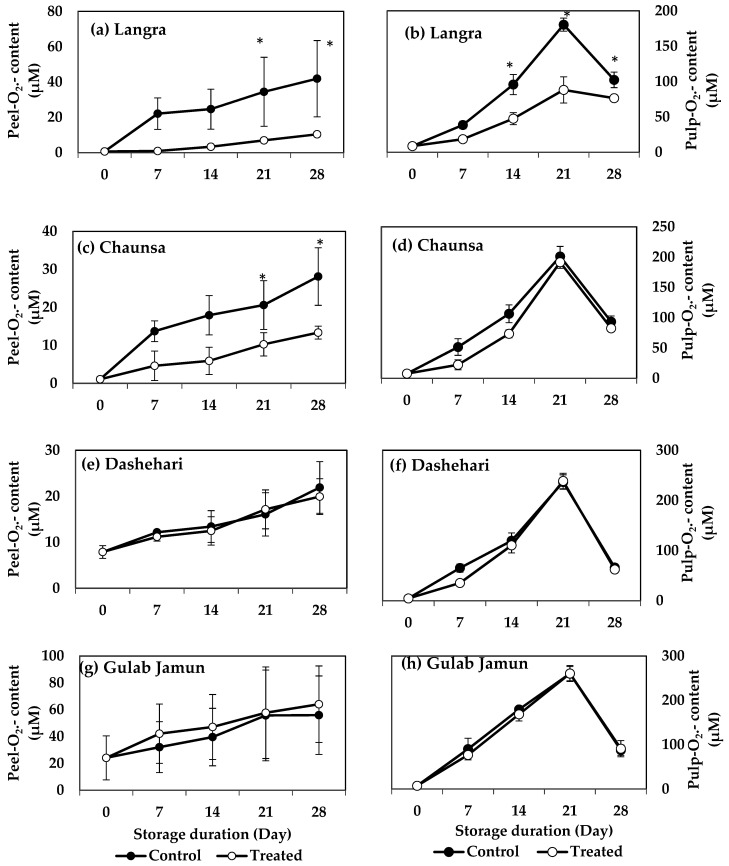
Effect of 100 µM melatonin treatment on O_2_^·−^ content in (**a**) ‘Langra’ peel, (**b**) ‘Langra’ pulp, (**c**) ‘Chaunsa’ peel, (**d**) ‘Chaunsa’ pulp, (**e**) ‘Dashehari’ peel, (**f**) ‘Dashehari’ pulp, (**g**) ‘Gulab Jamun’ peel, and (**h**) ‘Gulab Jamun’ pulp during storage at 5 ± 1 °C plus 3 d of shelf life at room temperature. Data are the mean of three replicates ± standard error. An asterisk (*) on the same storage period indicates significant differences (*p* < 0.05).

**Figure 3 antioxidants-11-00384-f003:**
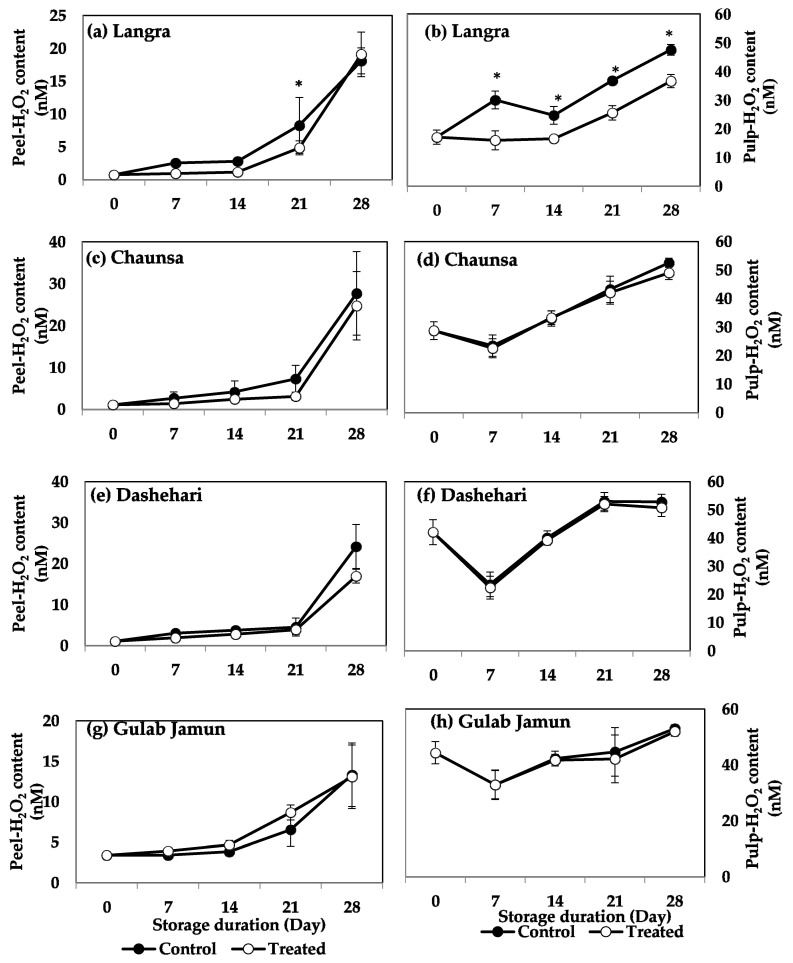
H_2_O_2_ content in (**a**) ‘Langra’ peel, (**b**) ‘Langra’ pulp, (**c**) ‘Chaunsa’ peel, (**d**) ‘Chaunsa’ pulp, (**e**) ‘Dashehari’ peel, (**f**) ‘Dashehari’ pulp, (**g**) ‘Gulab Jamun’ peel, and (**h**) ‘Gulab Jamun’ pulp as affected by 100 µM melatonin (MT) treatment during storage at 5 ± 1 °C plus 3 d of shelf life at room temperature. Data are the mean of three replicates ± standard error. An asterisk (*) on the same storage period indicates significant differences (*p* < 0.05).

**Figure 4 antioxidants-11-00384-f004:**
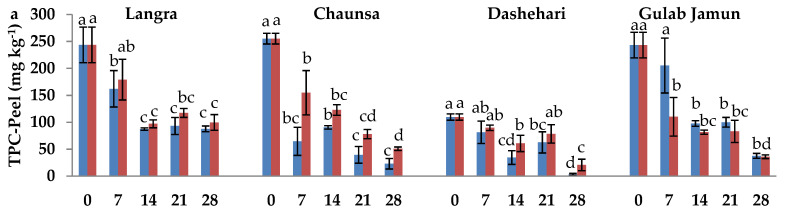
Total phenolic content (TPC) (**a**) in peel and (**b**) in pulp, and total flavonoid content (TFC) (**c**) in peel and (**d**) in pulp of ‘Langra’, ‘Chaunsa’, ‘Dashehari’, and ‘Gulab Jamun’ mangoes as affected by 100 µM melatonin (MT) treatment during storage at 5 ± 1 °C plus 3 d of shelf life at room temperature. Data are the mean of three replicates ± standard error. Different lowercase letters on the same storage period indicate significant differences (*p* < 0.05).

**Figure 5 antioxidants-11-00384-f005:**
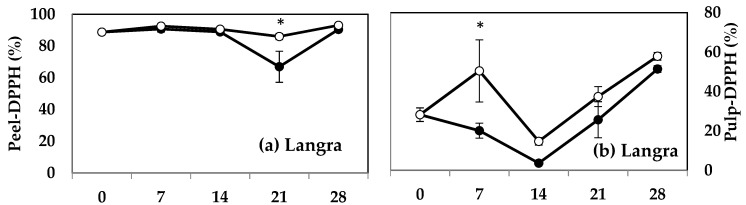
DPPH assay in (**a**) ‘Langra’ peel, (**b**) ‘Langra’ pulp, (**c**) ‘Chaunsa’ peel, (**d**) ‘Chaunsa’ pulp, (**e**) ‘Dashehari’ peel, (**f**) ‘Dashehari’ pulp, (**g**) ‘Gulab Jamun’ peel, and (**h**) ‘Gulab Jamun’ pulp as affected by 100 µM melatonin (MT) treatment during storage at 5 ± 1 °C plus 3 d of shelf life at room temperature. Data are the mean of three replicates ± standard error. An asterisk (*) on the same storage period indicates significant differences (*p* < 0.05).

**Figure 6 antioxidants-11-00384-f006:**
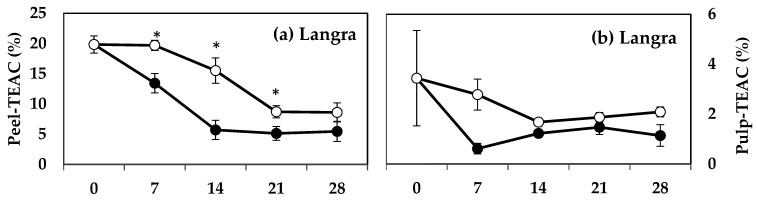
Effect of 100 µM melatonin (MT) treatment on Trolox equivalent antioxidant capacity (TEAC) assay in (**a**) ‘Langra’ peel, (**b**) ‘Langra’ pulp, (**c**) ‘Chaunsa’ peel, (**d**) ‘Chaunsa’ pulp, (**e**) ‘Dashehari’ peel, (**f**) ‘Dashehari’ pulp, (**g**) ‘Gulab Jamun’ peel, and (**h**) ‘Gulab Jamun’ pulp t during storage at 5 ± 1 °C plus 3 d of shelf life at room temperature. Data are the mean of three replicates ± standard error. An asterisk (*) on the same storage period indicates significant differences (*p* < 0.05).

**Figure 7 antioxidants-11-00384-f007:**
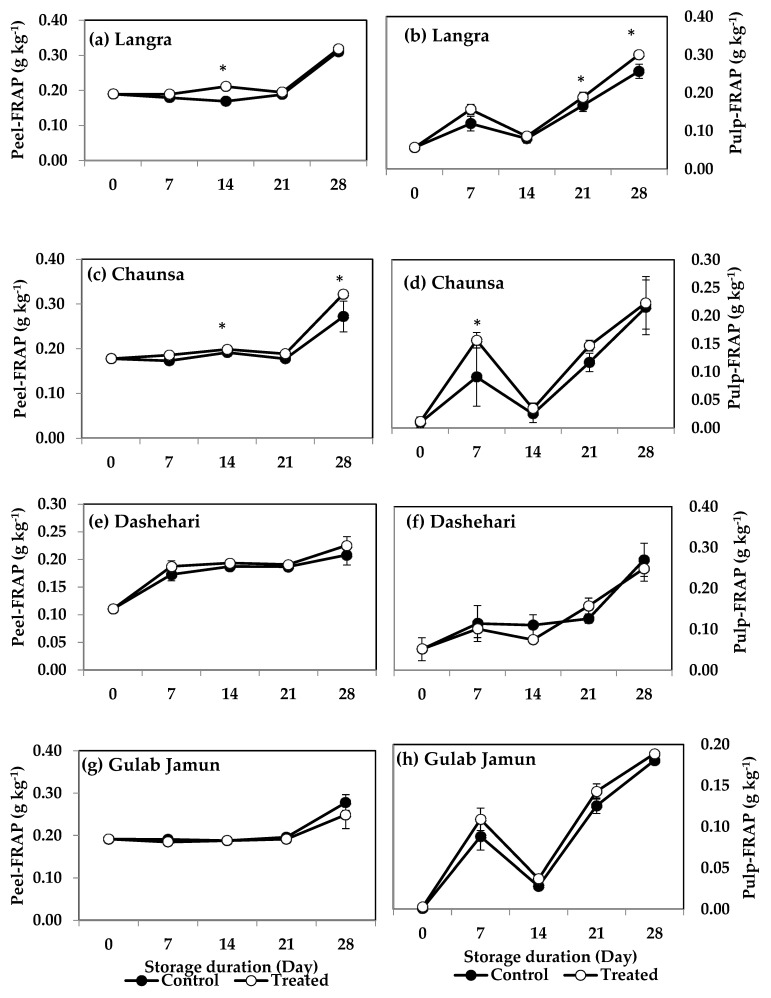
Ferric-reducing antioxidant power (FRAP) in (**a**) ‘Langra’ peel, (**b**) ‘Langra’ pulp, (**c**) ‘Chaunsa’ peel, (**d**) ‘Chaunsa’ pulp, (**e**) ‘Dashehari’ peel, (**f**) ‘Dashehari’ pulp, (**g**) ‘Gulab Jamun’ peel, and (**h**) ‘Gulab Jamun’ pulp as affected by 100 µM melatonin (MT) treatment during storage at 5 ± 1 °C plus 3 d of shelf life at room temperature. Data are the mean of three replicates ± standard error. An asterisk (*) on the same storage period indicates significant differences (*p* < 0.05).

**Figure 8 antioxidants-11-00384-f008:**
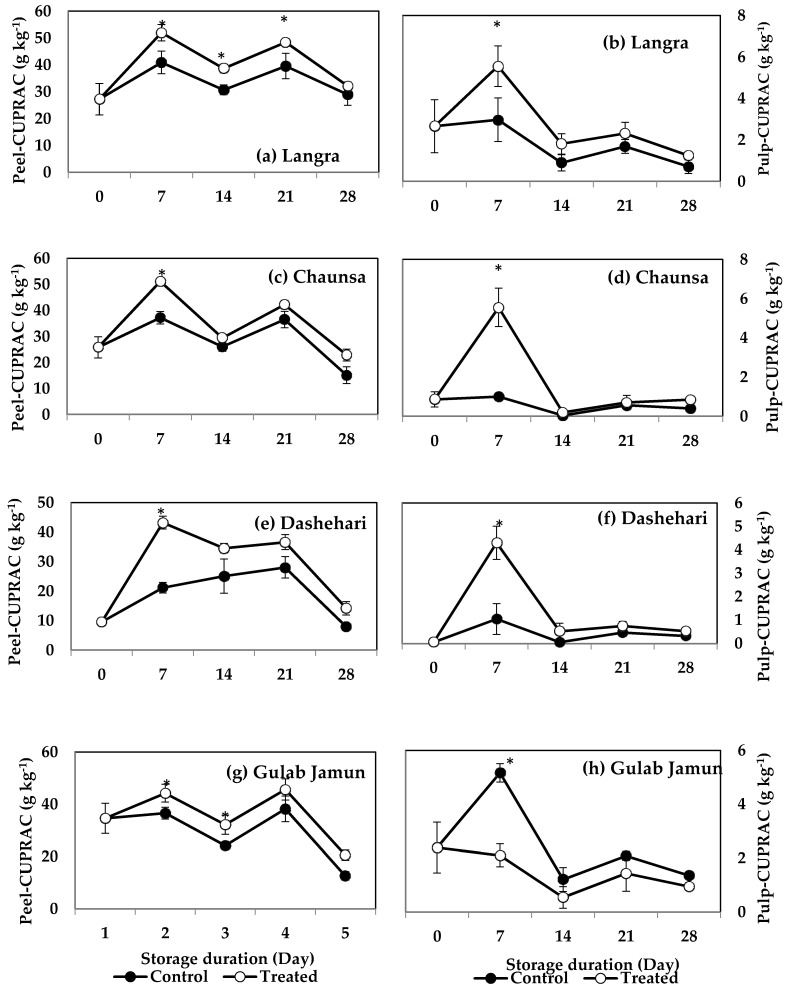
Cupric-reducing antioxidant power (CUPRAC) assay in (**a**) ‘Langra’ peel, (**b**) ‘Langra’ pulp, (**c**) ‘Chaunsa’ peel, (**d**) ‘Chaunsa’ pulp, (**e**) ‘Dashehari’ peel, (**f**) ‘Dashehari’ pulp, (**g**) ‘Gulab Jamun’ peel, and (**h**) ‘Gulab Jamun’ pulp as affected by 100 µM melatonin (MT) treatment during storage at 5 ± 1 °C plus 3 d of shelf life at room temperature. Dta are the mean of three replicates ± standard error. An asterisk (*) on the same storage period indicates significant differences (*p* < 0.05).

**Figure 9 antioxidants-11-00384-f009:**
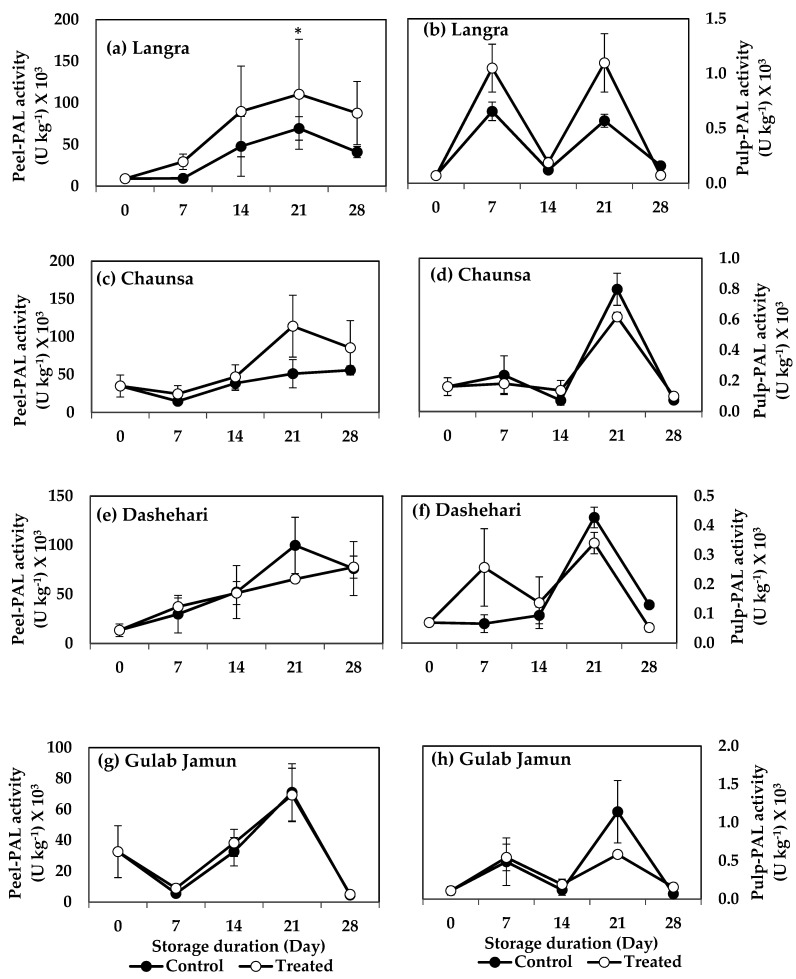
Effect of 100 µM melatonin treatment on phenylalanine ammonia lyase (PAL) activity in (**a**) ‘Langra’ peel, (**b**) ‘Langra’ pulp, (**c**) ‘Chaunsa’ peel, (**d**) ‘Chaunsa’ pulp, (**e**) ‘Dashehari’ peel, (**f**) ‘Dashehari’ pulp, (**g**) ‘Gulab Jamun’ peel, and (**h**) ‘Gulab Jamun’ during storage at 5 ± 1 °C plus 3 d of shelf life at room temperature. Data are the mean of three replicates ± standard error. An asterisk (*) on the same storage period indicates significant differences (*p* < 0.05).

**Figure 10 antioxidants-11-00384-f010:**
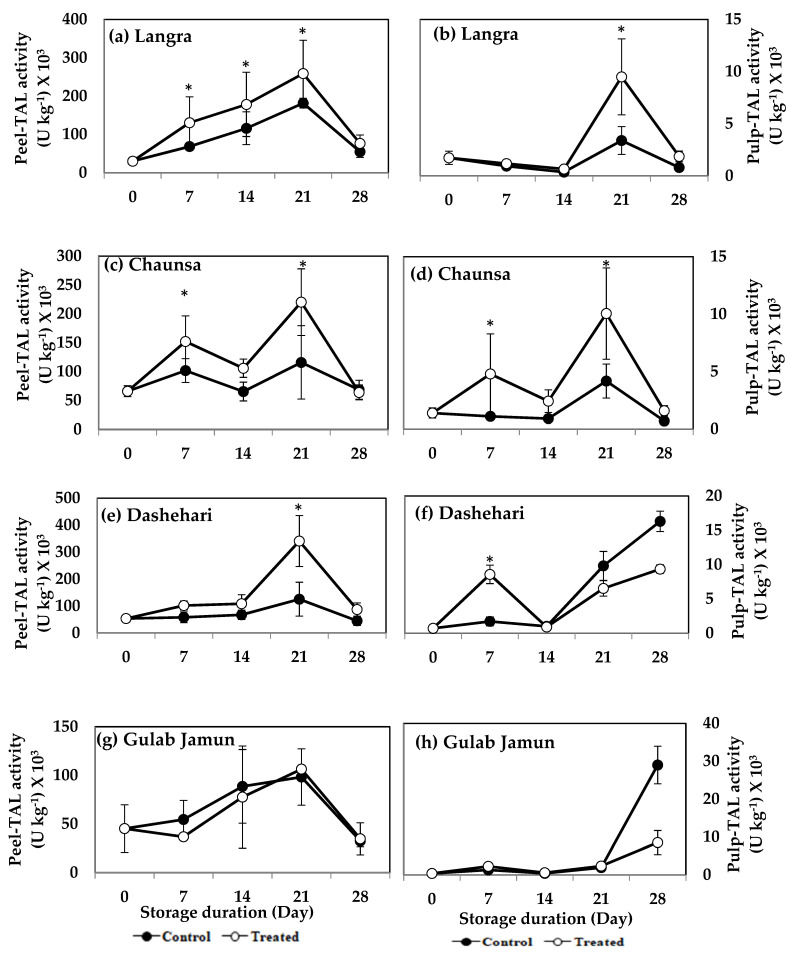
Effects of 100 µM melatonin treatment on tyrosine ammonia lyase (TAL) activity in (**a**) ‘Langra’ peel, (**b**) ‘Langra’ pulp, (**c**) ‘Chaunsa’ peel, (**d**) ‘Chaunsa’ pulp, (**e**) ‘Dashehari’ peel, (**f**) ‘Dashehari’ pulp, (**g**) ‘Gulab Jamun’ peel, and (**h**) ‘Gulab Jamun’ pulp during storage at 5 ± 1 °C plus 3 d of shelf life at room temperature. Data are the mean of three replicates ± standard error. An asterisk (*) on the same storage period indicates significant differences (*p* < 0.05).

**Figure 11 antioxidants-11-00384-f011:**
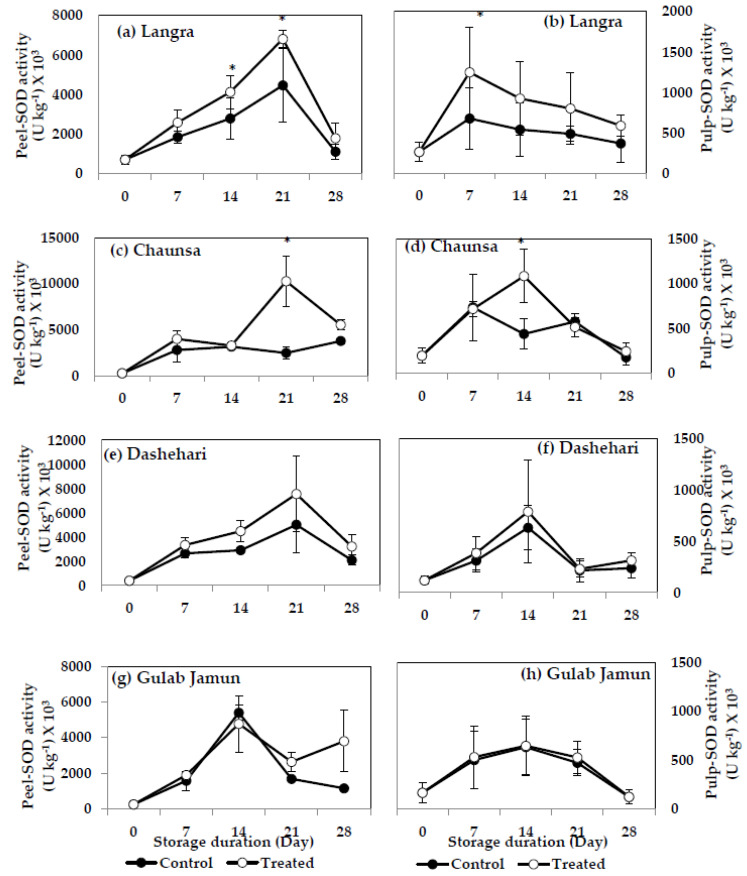
Superoxide dismutase (SOD) activity in (**a**) ‘Langra’ peel, (**b**) ‘Langra’ pulp, (**c**) ‘Chaunsa’ peel, (**d**) ‘Chaunsa’ pulp, (**e**) ‘Dashehari’ peel, (**f**) ‘Dashehari’ pulp, (**g**) ‘Gulab Jamun’ peel, and (**h**) ‘Gulab Jamun’ pulp treated with 0 µM (control) or 100 µM (treated) MT for 2 h, followed by 28 d of low temperature storage (5 ± 1 °C) and 3 d of shelf life at room temperature. Measurements were taken every 7 d of storage. Each value is the mean of three replicates ± standard error. An asterisk (*) on the same storage period indicates significant differences (*p* < 0.05).

**Figure 12 antioxidants-11-00384-f012:**
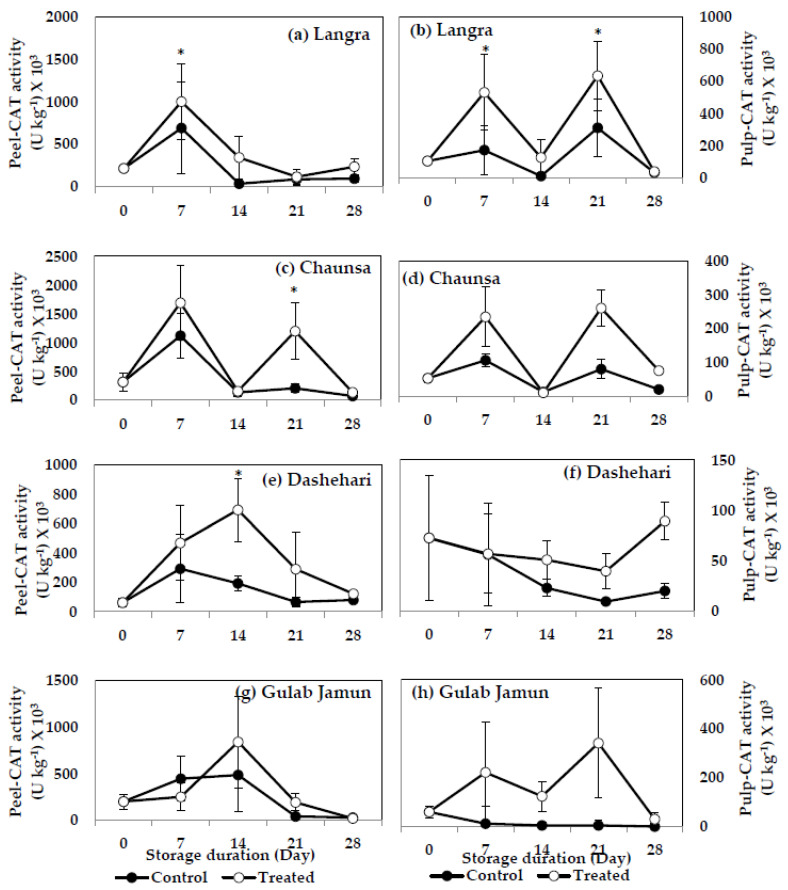
Catalase (CAT) activity in (**a**) ‘Langra’ peel, (**b**) ‘Langra’ pulp, (**c**) ‘Chaunsa’ peel, (**d**) ‘Chaunsa’ pulp, (**e**) ‘Dashehari’ peel, (**f**) ‘Dashehari’ pulp, (**g**) ‘Gulab Jamun’ peel, and (**h**) ‘Gulab Jamun’ pulp treated with 0 µM (control) or 100 µM (treated) MT for 2 h, followed by 28 d of low temperature storage (5 ± 1 °C) and 3 d of shelf life at room temperature. Measurements were taken every 7 d of storage. Each value is the mean of three replicates ± standard error. An asterisk (*) on the same storage period indicates significant differences (*p* < 0.05).

**Figure 13 antioxidants-11-00384-f013:**
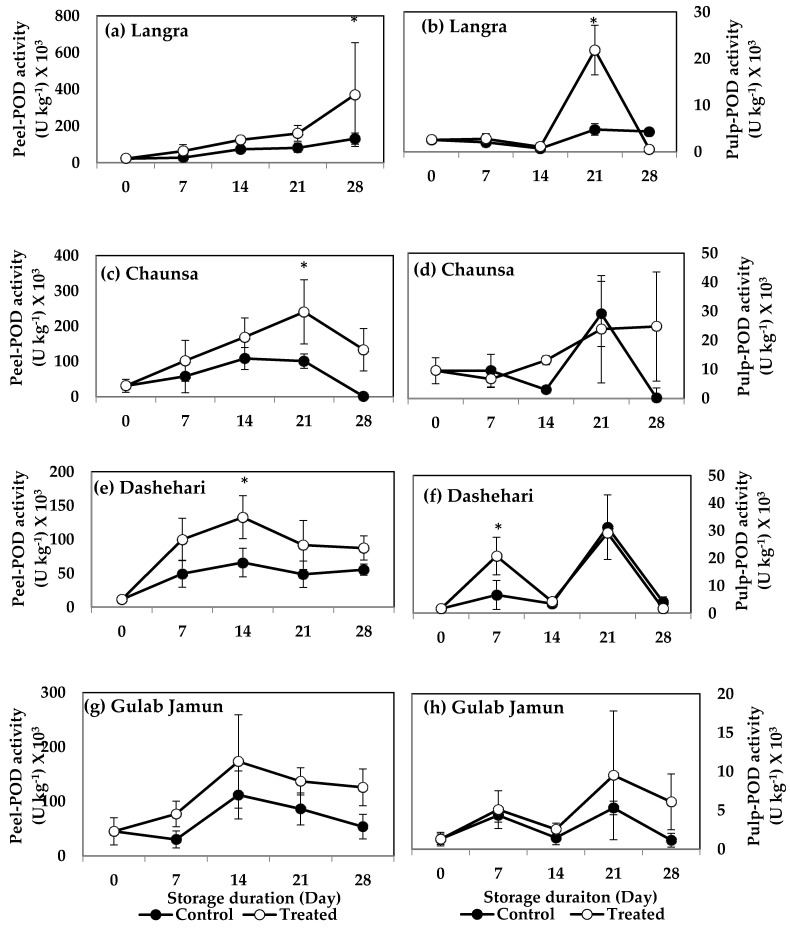
Effects of 100 µM melatonin treatment on peroxidase (POD) activity in (**a**) ‘Langra’ peel, (**b**) ‘Langra’ pulp, (**c**) ‘Chaunsa’ peel, (**d**) ‘Chaunsa’ pulp, (**e**) ‘Dashehari’ peel, (**f**) ‘Dashehari’ pulp, (**g**) ‘Gulab Jamun’ peel, and (**h**) ‘Gulab Jamun’ pulp during storage at 5 ± 1 °C plus 3 d of shelf life at room temperature. Data are the mean of three replicates ± standard error. An asterisk (*) on the same storage period indicates significant differences (*p* < 0.05).

**Figure 14 antioxidants-11-00384-f014:**
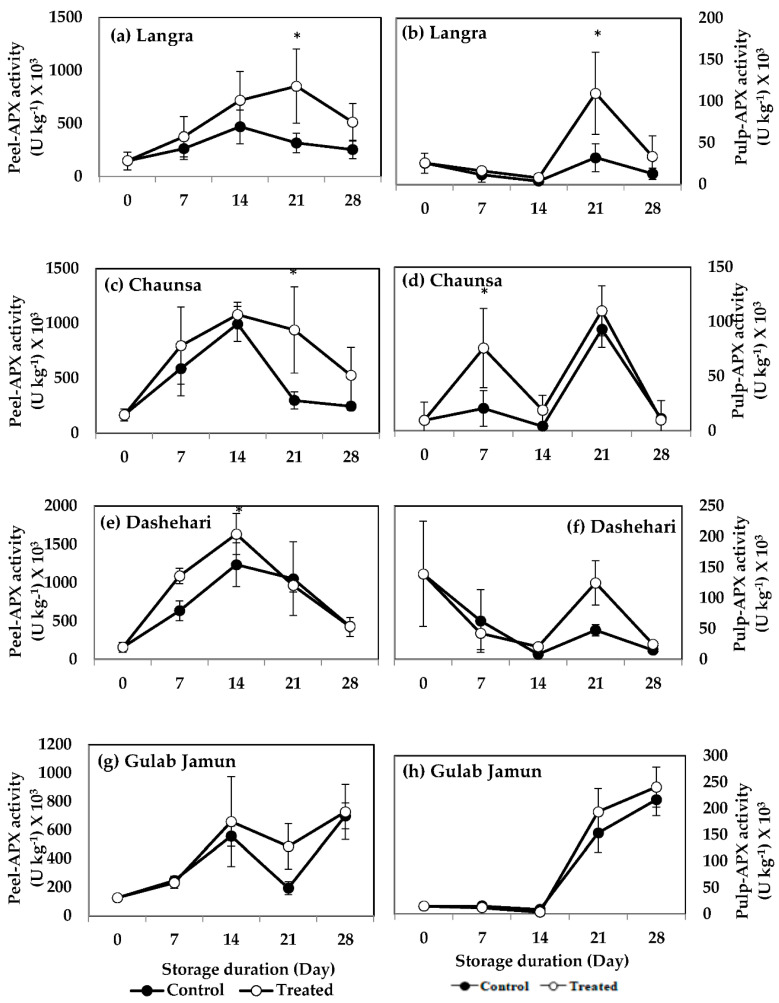
Effects of 100 µM melatonin treatment on ascorbate peroxidase (APX) activity in (**a**) ‘Langra’ peel, (**b**) ‘Langra’ pulp, (**c**) ‘Chaunsa’ peel, (**d**) ‘Chaunsa’ pulp, (**e**) ‘Dashehari’ peel, (**f**) ‘Dashehari’ pulp, (**g**) ‘Gulab Jamun’ peel, and (**h**) ‘Gulab Jamun’ pulp during storage at 5 ± 1 °C plus 3 d of shelf life at room temperature. Data are the mean of three replicates ± standard error. An asterisk (*) on the same storage period indicates significant differences (*p* < 0.05).

**Figure 15 antioxidants-11-00384-f015:**
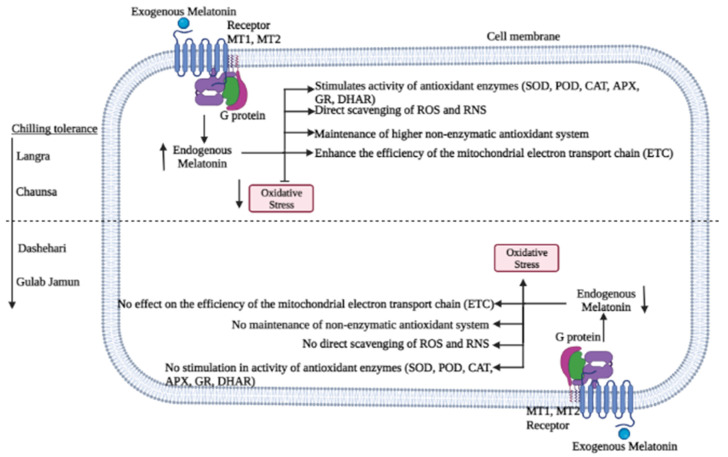
Possible mechanism of chilling injury alleviation in mango fruit by melatonin application.

## Data Availability

Data is contained within the article and Appendix A.

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
