# Peer review of "An Exogenous Pre-Storage Melatonin Alleviates Chilling Injury in Some Mango Fruit Cultivars, by Acting on the Enzymatic and Non-Enzymatic Antioxidant System"

_antioxidants, 2022, doi:10.3390/antiox11020384_

Round 1
Reviewer 1 Report
The article is interesting scientifically and useful in practice. The Spanish group is well known for the nice postharvest research on fruit quality after treatment with natural-phytochemical compounds.
There are some points, however, which have to be corrected or clarified:
-Line 22, please replace 'a higher' with 'an increased'\
-line 55, replace 'even while' with 'Although'
-line 75, please add phospholipids 'at low levels'. (The high FA saturation leads to CII).
-nM and mM are already indicate a concentration. Please, correct all the expressions of nM/kg, mM/kg etc properly.
-in graphs a) Please correct -1 to superscript, b) correct the X axis title, as in Figure 10 and anywhere else that cannot be seen clearly.
-Please, report whether you have run one-way or two-way analysis in M&M. Although I disagree with inserting the control numbers twice in a two-way ANOVA, it is generally accepted. This is just a comment.
-Line 501, replace 'intermediary' with 'intermediate' and anywhere else with a meaning similar to intermediate.
-Line 549, < augmenting the oxidative stress> or augmenting the stress responses??. Please clarify.
-Line 536, 'Similar to....' syntax error.
-In Discussion, please, discriminate whether each response was observed in the peel or the flesh or both.
-Did you observe a thicker peel in Gulab in comparison to other cultivars?
Author Response
Reviewer 1:
The article is interesting scientifically and useful in practice. The Spanish group is well known for the nice postharvest research on fruit quality after treatment with natural-phytochemical compounds. There are some points, however, which have to be corrected or clarified:
Our response:
Dear reviewer, thank you for your revision which has aided to improve our manuscript. We have made all the suggested changes.
Reviewer 1:
-Line 22, please replace 'a higher' with 'an increased'\
Our response:
The change was made as suggested.
Reviewer 1:
-line 55, replace 'even while' with 'Although'
Our response:
The change was made as requested.
Reviewer 1:
-line 75, please add phospholipids 'at low levels'. (The high FA saturation leads to CII).
Our response:
The change was made as requested.
Reviewer 1:
-nM and mM are already indicate a concentration. Please, correct all the expressions of nM/kg, mM/kg etc properly.
Our response:
Reviewer 1:
-in graphs a) Please correct -1 to superscript, b) correct the X axis title, as in Figure 10 and anywhere else that cannot be seen clearly.
Our response:
The “-1” was changed to “-1”. The axes and literals were aligned to avoid any overlapping text.
Reviewer 1:
-Please, report whether you have run one-way or two-way analysis in M&M. Although I disagree with inserting the control numbers twice in a two-way ANOVA, it is generally accepted. This is just a comment.
Our response:
We ran a one-way ANOVA. This has been specified in line 309.
Reviewer 1:
-Line 501, replace 'intermediary' with 'intermediate' and anywhere else with a meaning similar to intermediate.
Our response:
The term “intermediary” was changed to “intermediate” in lines 30, 517, 586, 594 and 628.
Reviewer 1:
-Line 549, < augmenting the oxidative stress> or augmenting the stress responses??. Please clarify.
Our response:
We meant to say that phenolics and flavonoids augment the fruits’ response to oxidative stress. This has been clarified in line 568.
Reviewer 1:
-Line 536, 'Similar to....' syntax error.
Our response:
This phrase was moved after the end of the following sentence, in order to clarify our intended meaning (lines 552-556).
Reviewer 1:
-In Discussion, please, discriminate whether each response was observed in the peel or the flesh or both.
Our response:
This issue ha been addressed in the revised manuscript according to your suggestion.
Reviewer 1:
-Did you observe a thicker peel in Gulab in comparison to other cultivars?
Our response:
Yes
Reviewer 2 Report
Title:An exogenous pre-storage melatonin alleviates chilling injury in some mango fruit cultivars, by acting on the enzymatic and non-enzymatic antioxidant system
Confidential Comments to the EIC
The author presented a study on the roles of MT application in improving low temperature stress tolerance in mango fruits. There were many published works about the action of MT as exogenous substance to increase plant stress tolerance. Here, the author determined endogenous MT content, ROS concentration, TPC content, TFC content, antioxidant enzyme activities, PAL activity, and TAL activity. The results suggested that exogenous MT alleviation of CI by increasing the content of endogenous MT, preventing the accumulation of ROS, and increasing antioxidant enzyme activities. The major results were clearly presented. However, there are some points needed to be revised. After the required revisions, I will recommend publishing.
*Comments to the Author:
Abstract:
The key points were highlighted, while there are some errors needed to be corrected.
L34: changes “CI” to “low temperature”
Introduction:
In this section, this section is fine and highlight objectives of this study.
Materials and methods:
This section is fine, the details of experimental methods have been well described.
Results:
In this section, the main results have been described clearly, while there are some errors needed to be corrected.
L356-369: What is the main conclusion from this section?
L370-387: What is the main conclusion from this section?
L415-491: What is the main conclusion from this section?
Discussion:
This section should be carefully improved. The role of MT decreased chilling injury under low temperature in mango fruit was not discussed well. In addition, some results should be supported by citing more references.
Zhao, Y.-Q. et al., 2021. Melatonin: A Potential Agent in Delaying Leaf Senescence. Crit. Rev. Plant Sci.: 1-22.
L522-523: Please rephrase this sentence, the logic is wrong.
L539-544: Please rephrase this sentence.
L570-571: Please delete this sentence.
Other comments: the format of figures should be consistent.
Author Response
REVIEWER 2
Confidential Comments to the EIC
The author presented a study on the roles of MT application in improving low temperature stress tolerance in mango fruits. There were many published works about the action of MT as exogenous substance to increase plant stress tolerance. Here, the author determined endogenous MT content, ROS concentration, TPC content, TFC content, antioxidant enzyme activities, PAL activity, and TAL activity. The results suggested that exogenous MT alleviation of CI by increasing the content of endogenous MT, preventing the accumulation of ROS, and increasing antioxidant enzyme activities. The major results were clearly presented. However, there are some points needed to be revised. After the required revisions, I will recommend publishing.
Our response:
Dear reviewer, thank you for your comments and suggestions.
Reviewer 2:
*Comments to the Author:
Abstract:
The key points were highlighted, while there are some errors needed to be corrected.
L34: changes “CI” to “low temperature”
Our response:
The term was changed as requested (lines 33-34).
Reviewer 2:
Introduction:
In this section, this section is fine and highlight objectives of this study.
Our response:
Thank you for your observation.
Reviewer 2:
Materials and methods:
This section is fine, the details of experimental methods have been well described.
Our response:
Thank you for your observation.
Reviewer 2:
Results:
In this section, the main results have been described clearly, while there are some errors needed to be corrected.
L356-369: What is the main conclusion from this section?
Our response:
Thank you for your observation. This section considers the effects of the MT treatment on total phenolics (TPC) and flavonoids (TFC), in both the peel and pulp of the fruits. Some significant changes were documented on either tissue, but most occurred on ‘Langra’, ‘Chaunsa’ and ‘Dashehari’ mangoes, with minimal (if at all) on ‘Gulab Jamun’ mangoes. This has been summarised in lines 371-374.
Reviewer 2:
L370-387: What is the main conclusion from this section?
Our response:
Antioxidant capacity was measured with four different methods (DPPH, TEAC, FRAP and CUPRAC). It was apparent that each method responded differently to each tissue and/or cultivar. This has been summarised in lines 393-397.
Reviewer 3:
L415-491: What is the main conclusion from this section?
Our response:
The effects of the MT treatment on the main biosynthesizing enzymes (PAL and TAL), were mostly exerted on PAL. Its effects on the activities of antioxidant enzymes (SOD, CAT and POD) were apparent on the peels of most cultivars, with the exception of ‘Gulab Jamun’ mangoes. This has been summarised in lines 435-437 and 461-463.
Reviewer 2:
Discussion:
This section should be carefully improved. The role of MT decreased chilling injury under low temperature in mango fruit was not discussed well. In addition, some results should be supported by citing more references.
Zhao, Y.-Q. et al., 2021. Melatonin: A Potential Agent in Delaying Leaf Senescence. Crit. Rev. Plant Sci.: 1-22.
Our response:
Thank you for your observation. The suggested citation is not directly related with our work.
Reviewer 2:
L522-523: Please rephrase this sentence, the logic is wrong.
L539-544: Please rephrase this sentence.
L570-571: Please delete this sentence.
Our response:
These phrases were edited (lines 538-540 and 556-564) or deleted (line 589) as recommended.
Reviewer 2:
Other comments: the format of figures should be consistent.
Our response:
The format of all figures has been revised (according to comments made by Reviewer 1 and our own changes), and homogenised.
Editor’s comments:
We noticed that the captions of the figures (highlighted part in the attachment) are similar to your previous publication (https://www.sciencedirect.com/science/article/abs/pii/S0925521421002416?via%3Dihub), please kindly revise the overlaps and explain the novelty of your research during major revision.
Our response:
Caption of the figures have been changed to avoid overlapping with the previous paper.
Round 2
Reviewer 1 Report
The authors have changed all required comments and the article is ready for further process.
Reviewer 2 Report
All comments have been considered, this version can be accepted.